# Fusion High-Resolution Network for Diagnosing ChestX-ray Images

**Zhiwei Huang** [1,2,†] , **Jinzhao Lin** [3,*] , **Liming Xu** [4,†] , **Huiqian Wang** [3] , **Tong Bai** [3] , **Yu Pang** [3,*]
**and Teen-Hang Meen** [5]

1    School of Communication and Information Engineering, Chongqing University of Posts and
Telecommunications, Chongqing 400065, China; hzwnet@swmu.edu.cn
2    School of Medical Information and Engineering, Southwest Medical University, Luzhou 646000, China
3    Chongqing Key Laboratory of Photoelectronic Information Sensing and Transmitting Technology,
Chongqing University of Posts and Telecommunications, Chongqing 400065, China;
wanghq@cqupt.edu.cn (H.W.); baitong03@126.com (T.B.)
4    Chongqing Key Laboratory of Image Cognition, School of Computer Science and Technology,
Chongqing University of Posts and Telecommunications, Chongqing 400065, China; xulimmail@gmail.com
5    Department of Electronic Engineering, National Formosa University, Yunlin 632, Taiwan;
thmeen@nfu.edu.tw
*   Correspondence: linjz@cqupt.edu.cn (J.L.); pangyu@cqupt.edu.cn (Y.P.)
†   These authors contributed equally to this work.

**Abstract:** The application of deep convolutional neural networks (CNN) in the field of medical image processing has attracted extensive attention and demonstrated remarkable progress. An increasing number of deep learning methods have been devoted to classifying ChestX-ray (CXR) images, and most of the existing deep learning methods are based on classic pretrained models, trained by global ChestX-ray images. In this paper, we are interested in diagnosing ChestX-ray images using our proposed Fusion High-Resolution Network (FHRNet). The FHRNet concatenates the global average pooling layers of the global and local feature extractors—it consists of three branch convolutional neural networks and is fine-tuned for thorax disease classification. Compared with the results of other available methods, our experimental results showed that the proposed model yields a better disease classification performance for the ChestX-ray 14 dataset, according to the receiver operating characteristic curve and area-under-the-curve score. An ablation study further confirmed the effectiveness of the global and local branch networks in improving the classification accuracy of thorax diseases.

**Keywords:** thorax disease classification; deep learning; ChestX-ray 14 dataset; feature fusion

## 1. Introduction

ChestX-rays (CXRs) are often included in routine physical examinations. Due to the advantages of being rapid, simple and economical, X-ray photography has become the most popular method for performing chest examinations [1]. A ChestX-ray can clearly record gross lesions of the lungs, including pneumonia, masses and nodules. The interpretation of CXR images in current medical practice, however, is mainly performed by radiologists, through artificial reading. The ChestX-ray image of a patient needs to be read by a senior radiologist for at least 10 min to make a diagnosis and different doctors can make inconsistent diagnoses of the same ChestX-ray image, which means that the results are affected by the cognitive ability of the radiologist, subjective experience, fatigue and other factors [2]. Computer-aided diagnosis (CAD) can overcome the deficiencies of radiologists, make

quick and effective objective judgements, improve accuracy and stability and reduce misdiagnoses and missed diagnoses [3–5].

Recently, benefitting from deep learning techniques [6], computer vision [7] has had remarkable success in the fields of target detection [8], image classification [9,10] and image inpainting [11], for example. This notable progress has led to the development of many medical image processing applications, including disease classification [12], lesion detection or segmentation [13–15], registration [16], image annotation [17,18] as well as other examples [19]. Deep learning methods, particularly deep convolutional neural networks (CNN) [20,21], have quickly become the preferred approach for processing medical images [22,23]. Large-scale datasets are usually required to train deep neural networks [24]. The ChestX-ray 14 dataset, released by the National Institutes of Health (NIH) in 2017 [25], is known as one of the largest hospital-scale ChestX-ray datasets. A series of studies was conducted to classify thoracic disease using this dataset. Existing CXR image diagnosis with deep learning [26–33] was used to resize or down-sample the high-resolution or original high-pixel images and eliminate most of the pixels in the images, with the hope that useful disease information would not be lost. The mainstream framework of a CNN for diagnosing thorax disease is shown in Figure 1, in which the input size of the CXR image is normally set to 224 × 224 × 3. For example, Mao et al. [34] used deep generative classifiers to make the model architecture more robust and to reduce model overfitting. Guan et al. [35] treated the entire image as a global branch, focused on local regions with disease specificity, and proposed an attention guided convolutional neural network (AG-CNN) to fuse complementary information for favourable accuracy. Zhu et al. [36] proposed the deep-local global feature fusion (DLGFF) framework, for multilevel semantic recognition in high spatial resolution images, which fused the local and global convolutional features and considered fully connected features. Lin et al. [37] set the outputs of a trained CNN as fuzzy integral inputs and proposed evolutionary-fuzzy-integral-based CNNs (EFI-CNN) for improved classification accuracy.

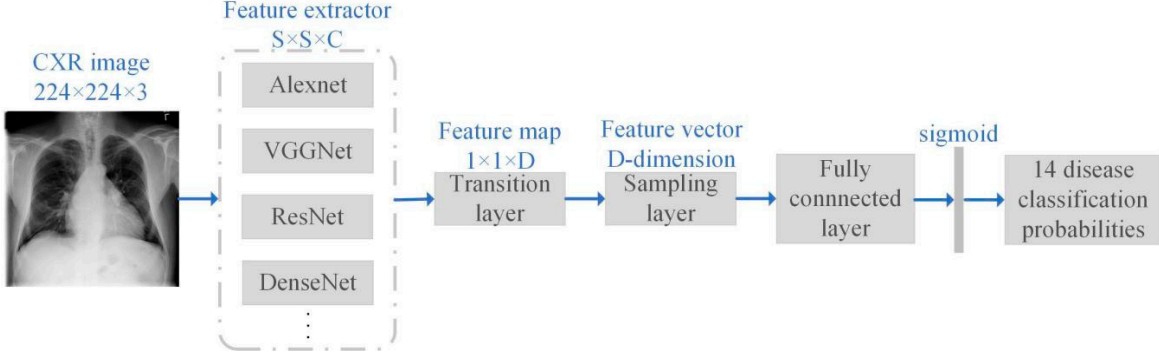

**Figure 1.** The mainstream framework of a convolutional neural network for diagnosing thorax disease.

The classic pretrained models, e.g., AlexNet [38], VGGNet [39], ResNet [40] and DenseNet [41], all use a CXR image that is resized to 224 × 224 × 3 as the input. The model encodes the image to C feature maps that are sized S × S and outputs them to the transition layer. Each feature map is reduced to 1 × 1 × D by the transition layer and then transformed into a D-dimensional feature vector by the sampling layer. A sigmoid function transforms the fully connected layer and then outputs probability scores for 14 thorax diseases.

Medical disease diagnosis, however, often needs to find abnormal disease information from dozens of pixels in a picture with millions of pixels to make an accurate disease judgement. Artificial downsampling, or discarding pixels, will result in the loss of disease information, missed diagnoses and misdiagnoses, leading to the treatment of the patient's diseases potentially being delayed.

In this paper, to take full advantage of neural network architectures and fuse image representation features, we adopt a fusion convolutional neural network and introduce the classification layer into a high-resolution network (HRNet) to improve the classification of CXR images. An illustration of HRNet

is provided in Figure 2. Specifically, four high-resolution feature maps are first fed into a bottleneck, and the number of output channels is increased to 64, 128, 256 and 512. The high-resolution representations are then downsampled by a 2-stride $3 \times 3$ convolution layer, which results in 128 channels. Then, all the channels are compiled into representations of the second-level high-resolution representations, and this process is conducted twice, to obtain 256 channels at the low resolution. Finally, the 512 channels are transformed into 1024 channels through one $1 \times 1$ convolution, which is followed by a global average pooling operation. The output 1024-dimensional representation is fed into the classifier [42].

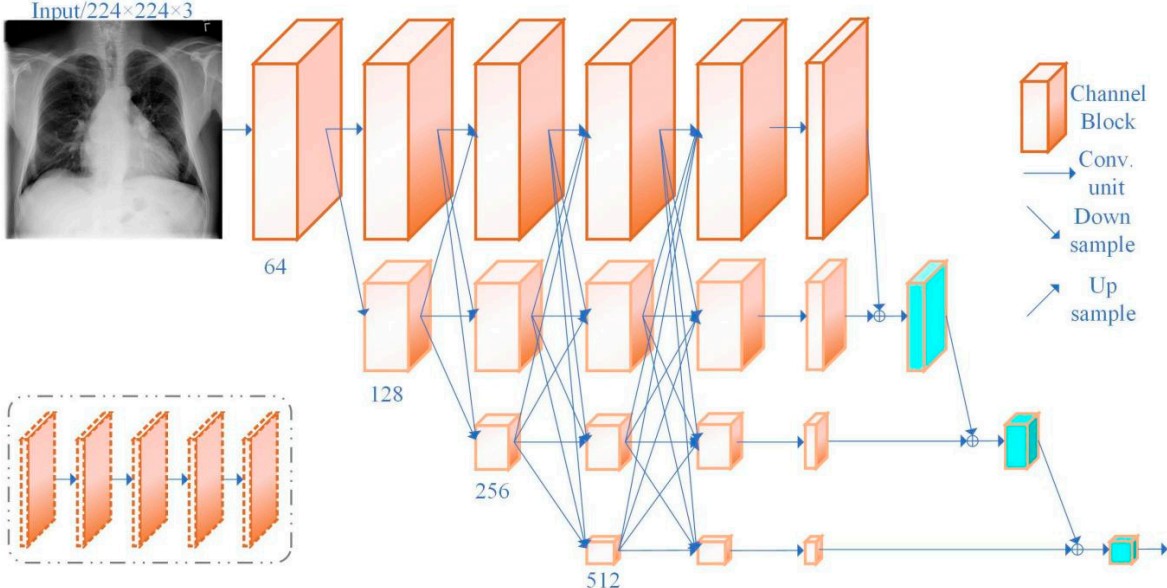

**Figure 2.** An architectural illustration of the proposed Fusion High-Resolution Network (FHRNet). The FHRNet is composed of four parallel high-to-low resolution subnetworks that repeatedly exchange information across multiresolution subnetworks. The vertical and horizontal directions correspond to the scale of the feature maps and the depth of the network, respectively.

In summary, our contributions in this work are as follows: First, we propose the fusion high-resolution network as a feature extractor, which produces competitive results compared with those of other advanced methods. Second, we introduce a fusion CNN that diagnoses ChestX-ray images by combining local and global cues. The FHRNet improves the performance of thorax disease classification by reducing the impact of noise and highlighting lung regions. Third, we conduct a comparative experiment based on the ChestX-ray 14 dataset. The classification results show that the FHRNet model achieves better performance than other available approaches.

## 2. Method

### 2.1. Dataset

Wang et al. [25] released the ChestX-ray 14 dataset in October 2017, and it is the largest available ChestX-ray dataset by far. The ChestX-ray14 dataset includes 112,120 CXR images, involving 30,805 patients. The pixel size of every CXR image is $1024 \times 1024$, and all images are saved in PNG format, with an 8-bit greyscale value. Every image is labelled with 14 different thorax diseases, with features extracted from radiologist reports. The ground truth data are mined and labelled through natural language processing (NLP) from patient diagnostic reports, and the label accuracy is estimated to be greater than ninety percent. Among the 112,120 ChestX-ray images, 51,708 images contained one or more diseases, and the remaining 60,412 images were considered normal and labelled "No Finding". An image example is shown in Figure 3.

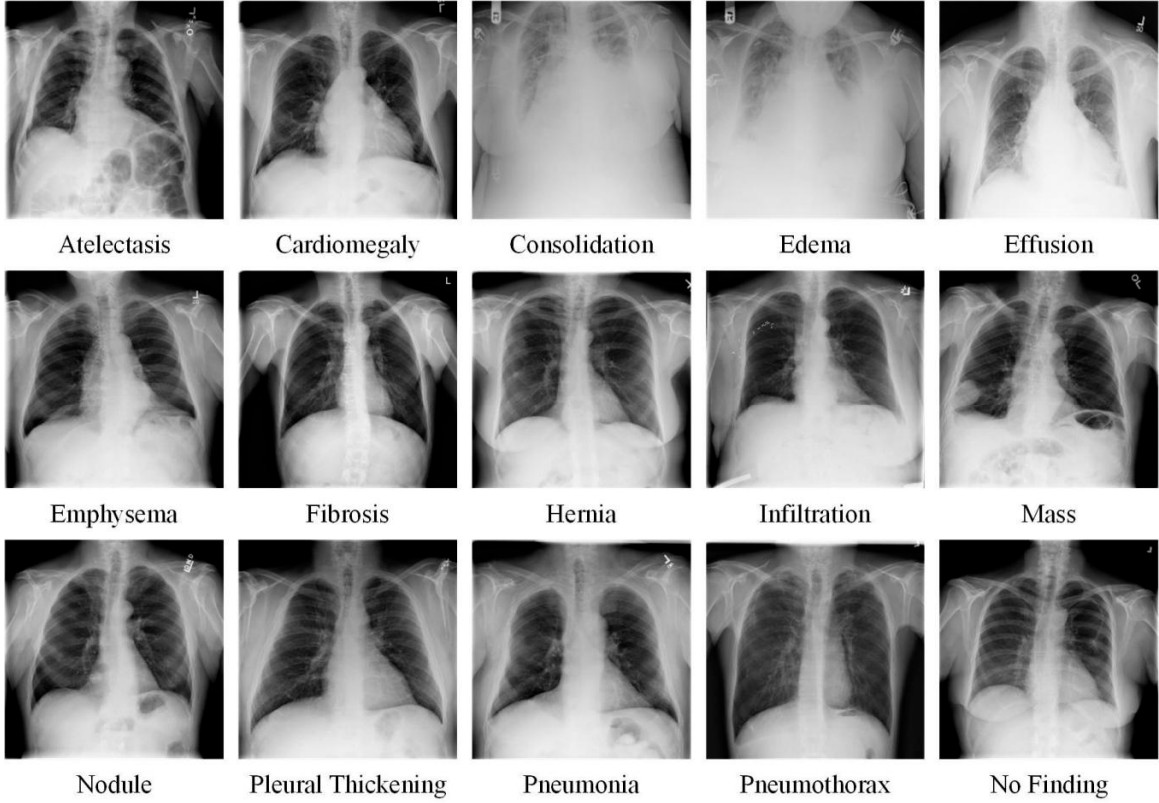

**Figure 3.** Example of images in the ChestX-ray 14 dataset.

The ChestX-ray 14 dataset includes multilabel classification and is large enough for deep learning; therefore, it was used to evaluate and validate the FHRNet model. In this experiment, we divided the whole dataset into a training set (total 75,708 images), a validation set (total 10,816 images) and a test set (total 25,596 images), at the hospital scale. All images from the same patient only appeared once in the training set, the validation set and the test set.

### 2.2. Network Framework

As shown in Figure 4, the proposed FHRNet has three branches: the local feature extractor, the global feature extractor and the feature fusion module. The local and global feature extractors are disease classification networks that output disease classification probabilities from the corresponding images. In contrast, the input image of the local feature extractor is a small lung region that is cropped using a mask inference generated from the global feature extractor. Two of the HRNets were adjusted to obtain the distinguishing features of the local lung region and whole image.

The HRNets are connected to global average pooling layers, a fully connected layer, a sigmoid layer and a loss function. The feature fusion module concatenates the global average pooling layers after two feature extraction steps and is then fine-tuned to make a final classification prediction.

### 2.3. Network Structure

It usually takes three steps to build a model for classifying CXR images, based on the deep learning of multibranch images. These steps are feature extraction, feature fusion and classification prediction. The specific descriptions of these steps are provided below.

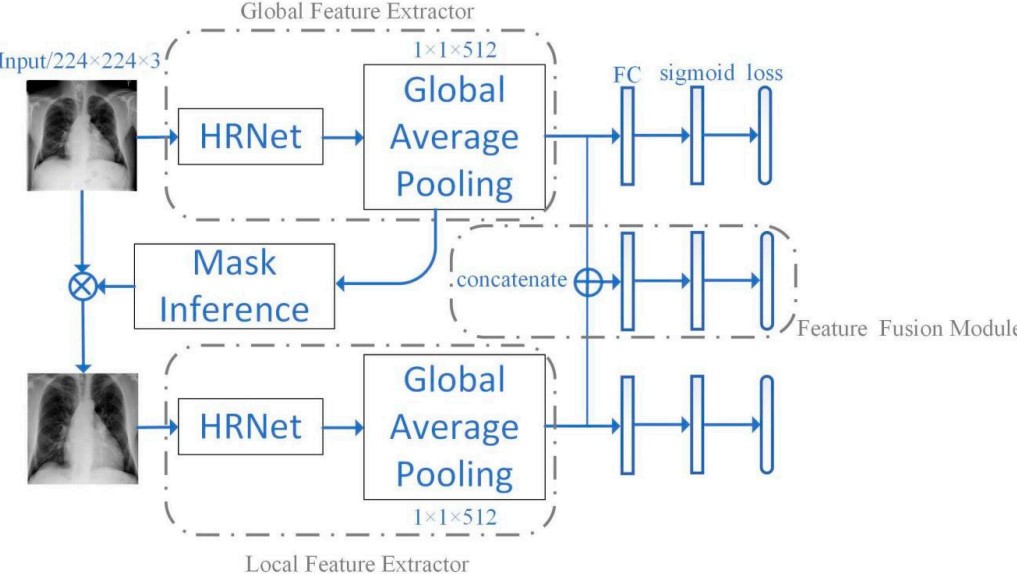

**Figure 4.** The overall framework of the FHRNet.

**Feature extraction from multibranch images.** Determining how to better extract features from multiview medical images is one of the main research topics in the field of medical image processing based on deep learning methods [43]. Although a variety of advantageous features have been manually extracted from multiview medical images, for example, HOG, LBP and SIFT, classification predictions based on these features can lead to incompatibility problems, that is, the extracted features cannot effectively classify and predict specific organs or diseases. Feature extraction based on a CNN solves the above problems. With the continuous development of attention mechanisms, feature extraction from multiview medical images has become increasingly ideal [44].

When we take the feature extraction network $f$ as an example, HRNet can be used to extract features. Suppose that the network can be expressed as follows:

$$f(x, \theta) = W^L a_{(L-1)}(W^{(L-1)} a_{(L-2)}(W^1 x + b^1) + b^{(L-1)}) + b^L \tag{1}$$

in which $\theta := \left\{ W^1, b^1, \ldots, W^{L-1}, b^{L-1}, W^L, b^L \right\}$ are the parameters of network $f$, $a_l (1 \le l < L)$ represents the activation function of the $l$th layer, $x$ represents the input of the network $f$. $f(x, \theta)$ represents an output that is not processed by the activation function of the last layer [45]. The overall output of the network is as follows:

$$Output = A(f(x, \theta)) \tag{2}$$

in which $A$ represents the activation function of feature extraction network $f$ [46].

As shown in Figure 2, the input of feature extraction network $f$ includes the global input image $x_g$ and the local input image $x_l$, and the $i$th local input image is represented as $x_l^i = m^i \odot x_g^i$. Therefore, according to the definition of the feature network, the global features and local features can be expressed as follows:

$$O_g = A(f_g(x_g, \theta)) \tag{3}$$

$$O_l = A\big(f_l\big(x_g \odot m, \theta\big)\big) \tag{4}$$

**Feature fusion from multibranch images.** To use the images of different branches for classification prediction, it is necessary to construct unified fusion features to share the features of different branches. After different deep neural networks extract the features from different branch images, the shared fusion features can be obtained by directly concatenating the images from the three branches,

$$O = w_1 \cdot A(f_g(x_g, \theta)) + w_2 \cdot A(f_l(x_l, \theta)) \tag{5}$$

in which $w_i(1 \leq i \leq 2)$ represents the weight of a feature that is extracted from the *i*th network of the fusion feature [47].

It is not difficult to find that the features extracted from the three branches will result in feature redundancy. An attention mechanism can be used to reduce feature redundancy. That is, adding a random mask after the last activation layer and removing redundant features can increase the classification accuracy.

**Classification prediction.** At present, the prediction of lung disease is a multiclassification task that usually adopts the softmax classification function. The classification function is expressed as

$$[p_1, p_2, \ldots, p_{14}] = Sofmax(W \times F) \tag{6}$$

in which $O$ represents the fusion feature, $W$ represents the mapping matrix that is used to map the high-dimensional fusion feature to a low-dimensional probability distribution representing the disease information and $p_i(1 \leq i \leq 14)$ represents the probability of identifying the *i*th disease [48].

To dynamically determine the weights of the three features and further improve the prediction accuracy, global and local consistency classification methods can be used. That is, three classifiers for global, local and fusion features are trained and alternately optimised for classification prediction,

$$[p_1^1, p_2^1, \ldots, p_{14}^1] = Sofmax(W \times O_g) \tag{7}$$

$$[p_1^2, p_2^2, \ldots, p_{14}^2] = Sofmax(W \times O_l) \tag{8}$$

$$[p_1^3, p_2^3, \ldots, p_{14}^3] = Sofmax(W \times O) \tag{9}$$

in which $p_i^j(1 \leq i \leq 14, 1 \leq j \leq 3)$ represents the probability that the *j*th network predicts the *i*th disease. According to the mechanism of global–local consistency, the probabilities of patients suffering from the 14 diseases are $p_1^1 p_1^2, p_2^1 p_2^2, \ldots$, and $p_{14}^1 p_{14}^2$. Due to the range of the probability values, the final diagnosis probability is small and it is processed by a logarithmic function to become useful to doctors.

## 3. Experimental Setting

In all pretrained models, input images are expected to be normalised by the same means, such as by creating a minibatch of three-channel RGB images ($3 \times H \times W$), in which either H or W is expected to be no less than 224. All images in the ChestX-ray 14 dataset are $1024 \times 1024$, with an 8-bit greyscale value. We split the dataset into the training set (78,468 images of 21,528 patients), the validation set (11,219 images of 3090 patients) and the test set (22,433 images of 6187 patients), without the same patient overlapping among sets. We converted these greyscale images to three-channel RGB images, cropped them to a $224 \times 224$ resolution at the centre and then normalised these images by the means ([0.485, 0.456, 0.406]) and standard deviations ([0.229, 0.224, 0.225]). We trained the model by the Adam optimiser and set the initial learning rate and batch size as $1.0 \times 10^{-4}$ and 32, respectively. We completed the training procedure after 50 epochs. After each epoch, we validated, tested and saved the model with the best classification performance. For multiclass classification, we used the receiver operating characteristic (ROC) curve and area-under-the-curve (AUC) score to assess the classification performance. The model weights associated with the best AUC scores, based on the validation set, were saved and used to extract representative features. In our experiment, we plotted the ROC curve for each thorax disease and calculated the AUC scores for 14 diseases to evaluate the classification performance. The FHRNet was implemented with the Pytorch 1.0 framework in Python 3.6 on an Ubuntu 16.04 server. The model was trained, validated and tested on an 8-core CPU and four TITAN V GPUs.

## 4. Results

The classification results for the existing methods and the FHRNet based on the ChestX-ray 14 dataset are presented in terms of the AUC scores in Table 1. The obtained ROC curves of the FHRNet for each of 14 thorax diseases are shown in Figure 5.

**Table 1.** The area-under-the-curve (AUC) scores of existing methods and the FHRNet based on the ChestX-ray 14 dataset. The scores that displayed a relative increase are marked in bold.

| Thorax Disease | Wang [25] | Yao [49] | Gundel [50] | FHRNet |
|---|---|---|---|---|
| Atelectasis | 0.7003 | 0.733 | 0.767 | **0.794** |
| Cardiomegaly | 0.8100 | 0.856 | 0.883 | **0.902** |
| Effusion | 0.7585 | 0.806 | 0.828 | **0.839** |
| Infiltration | 0.6614 | 0.673 | 0.709 | **0.714** |
| Mass | 0.6933 | 0.718 | 0.821 | **0.827** |
| Nodule | 0.6687 | 0.777 | 0.758 | 0.727 |
| Pneumonia | 0.6580 | 0.684 | 0.731 | 0.703 |
| Pneumothorax | 0.7993 | 0.805 | 0.846 | **0.848** |
| Consolidation | 0.7032 | 0.711 | 0.745 | **0.773** |
| Edema | 0.8052 | 0.806 | 0.835 | 0.834 |
| Emphysema | 0.8330 | 0.842 | 0.895 | **0.911** |
| Fibrosis | 0.7859 | 0.743 | 0.818 | **0.824** |
| Pleural Thickening | 0.6835 | 0.724 | 0.761 | 0.752 |
| Hernia | 0.8717 | 0.775 | 0.896 | **0.916** |
| Average | 0.7451 | 0.761 | 0.807 | **0.812** |

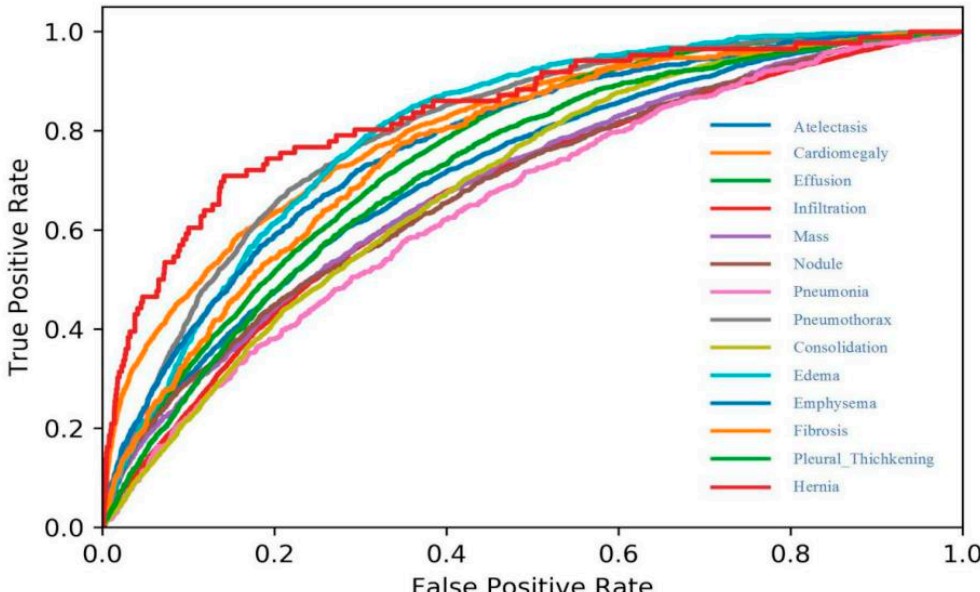

**Figure 5.** The receiver operating characteristic (ROC) curves of the FHRNet, for each of the 14 thorax diseases.

Based on the available published works performed by other researchers, including Wang [25], Yao [49] and Gundel [50], we recorded and compared the AUC scores they obtained and those of the FHRNet based on the ChestX-ray 14 dataset. We found that the FHRNet method achieved the expected effect and provided a superior classification performance. A numerical comparison of the results for 14 classes of thorax diseases and the average AUC of each method are shown in Table 1. Compared with the three existing methods, the proposed method increased the average AUC by 8.98% (from 0.7451 to 0.812). Notably, for "Mass", the rate of increase in the AUC score reached 19.3% (from 0.6933 to 0.827).

From Table 1, a horizontal comparison shows that the existing methods and our model obtained different classification effects, even for the same thorax disease. Among the 14 diseases, 10 thorax diseases had the best average AUC scores with the FHRNet model: "Atelectasis", "Cardiomegaly", "Effusion", "Infiltration", "Mass", "Pneumothorax", "Consolidation", "Emphysema", "Fibrosis" and "Hernia". Table 1 also shows that the FHRNet model achieved the best average AUC score.

A vertical comparison shows that the existing methods and our model obtain different classification effects for the 14 thorax diseases. The most accurately identified thorax disease was "Hernia", with an AUC score of 0.916, and the least accurately identified disease was "Pneumonia", with an AUC score of 0.703.

We also plotted the ROC curves of the FHRNet for each of the 14 thorax diseases, as shown in Figure 5. We can observe that the ROC curve of "Infiltration" was flatter than that of "Hernia", which means that the classification of "Pneumonia" was not as good as that of "Hernia".

## 5. Discussion

The experimental results show that the proposed FHRNet provides excellent disease classification performance. Our method can obtain satisfactory results because two significant structures are introduced: (1) a high-resolution network is adopted as a feature extractor to exchange image representation features and (2) the local and global branches of the ChestX-ray images are introduced to obtain the most useful features. To illustrate the effectiveness of local and global branches in our method, we conducted a further ablation study that correspondingly yielded different AUC scores. The results of the ablation study of local and global branches are shown in Table 2.

**Table 2.** The ablation study of local and global branches.

| Thorax Disease | Global Fusion | Local Fusion | FHRNet |
|---|---|---|---|
| Atelectasis | 0.778 | 0.783 | 0.794 |
| Cardiomegaly | 0.879 | 0.894 | 0.902 |
| Effusion | 0.822 | 0.828 | 0.839 |
| Infiltration | 0.703 | 0.697 | 0.714 |
| Mass | 0.804 | 0.816 | 0.827 |
| Nodule | 0.708 | 0.721 | 0.727 |
| Pneumonia | 0.684 | 0.692 | 0.703 |
| Pneumothorax | 0.836 | 0.844 | 0.848 |
| Consolidation | 0.758 | 0.764 | 0.773 |
| Edema | 0.827 | 0.821 | 0.834 |
| Emphysema | 0.897 | 0.903 | 0.911 |
| Fibrosis | 0.815 | 0.813 | 0.824 |
| Pleural Thickening | 0.735 | 0.453 | 0.752 |
| Hernia | 0.904 | 0.908 | 0.916 |
| Average | 0.803 | 0.806 | 0.812 |

We developed a three-branch convolutional neural network for diagnosing CXR images in this study. The fusion branch used two high-resolution networks to adaptively concentrate on pathologically abnormal regions, which thus improved the classification accuracy. The model achieved the effective utilisation of the fusion features extracted from both local lung region images and entire ChestX-ray images. If the fusion branch were to be eliminated, the performance of the FHRNet model would degrade. With reasonable confidence, we conclude that the fusion branch plays an important role in the FHRNet model. Among the existing methods that were trained only on the ChestX-ray 14 dataset, the FHRNet achieved good AUC scores for the 14 thorax diseases.

## 6. Conclusions

In this work, an innovative architecture, termed the FHRNet, was applied to classify 14 thorax diseases and diagnose ChestX-ray images. Compared with most previous networks, the difference is that the FHRNet consists of four parallel high-to-low resolution subnetworks and repeatedly exchanges information via multiscale fusion processes. Two HRNets were trained by the local and global feature extraction branches, and the feature fusion module was concatenated and fine-tuned for the final prediction. Our experimental results for the ChestX-ray14 dataset demonstrated the effectiveness and accuracy of the FHRNet model. Additional ablation studies showed that the local and global feature extraction branches affect the classification performance and improve the classification effect after fusion.

In our future work, we will focus on the pixel-level segmentation of the lung region, from CXR images, to further improve the classification performance. Then, we will train the model by using more than 180,000 images from the PLCO dataset [51] as extra training data for applying the model in computer-aided diagnosis.

**Author Contributions:** Z.H. designed and implemented the prototype, executed the experiments, analysed the results and wrote the article. L.X. analyzed the process of auxiliary diagnosis from deep learning view and executed the experiments. H.W. and T.B. performed the literature review. J.L., Y.P. and T.-H.M. performed grammar, logical structure and typo correction. All authors have read and agreed to the published version of the manuscript.

**Funding:** This work was funded by National Natural Science Foundation of China (61671091, 61971079), by Science and Technology Research Program of Chongqing Municipal Education Commission (KJQN201800614), by Chongqing Research Program of Basic Research and Frontier Technology (cstc2017jcyjBX0057, cstc2017jcyjAX0328), by Scientific Research Foundation of CQUPT(A2016-73), by Key Research Project of Sichuan Provincial Department of Education (18ZA0514) and by Joint Project of LZSTB-SWMU(2015LZCYD-S08(1/5)).

**Conflicts of Interest:** The authors declare no conflict of interest.

## Abbreviations

In this manuscript the used abbreviations are as follows:

FHRNet      Fusion High-Resolution Net
CXR         ChestX-ray
NLP         Natural Language Processing
CAD         Computer-Aided Diagnosis
CNN         Convolution Neural Network
NIH         National Institutes of Health
ROC         Receiver Operating Characteristic
AGCNN       Attention Guided Convolution Neural Network
AUC         Area Under Curve
LBP         Local Binary Pattern
HOG         Histogram of Oriented Gradients
SIFT        Scale-Invariant Feature Transform

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
