# Peer review of "Fusion High-Resolution Network for Diagnosing ChestX-ray Images"

_electronics, doi:10.3390/electronics9010190_

Round 1

Reviewer 1 Report

The paper proposes Fusion High Resolution Network in diagnosing chest X ray images, which performs well compared to existing CNN methods. In general, the manuscript gives a good description of the proposed methodology and results and conclusions are supported by the empirical data analyzed in the study. However, I believe the manuscript needs to be revised. The quality of the collected data is very good but I think the analysis presented in the manuscript is somehow limited and needs to be extended. The introduction part should be improved so that the relation to "electronic" is highlighted.

Author Response

Journal: Electronics (ISSN 2079-9292)

Manuscript ID: electronics-678921

Type: Article   Number of Pages: 9

Title: Fusion High-Resolution Network for Diagnosing Chest X-ray Images

Dear Editor,

Thank you very much for your letter and for the comments by the reviewers. These comments are very valuable and helpful for our paper.

We appreciate the careful, constructive, and generally favorable reviews given to our paper by the reviewers.

We believe we have adequately addressed all the excellent advices and questions raised by reviewers. Furthermore, we checked the manuscript and made sure the submitted manuscript is correct.

Please contact us if any further questions remain.

Sincerely yours,

Prof.  Yu Pang 

Response to the comments of reviewers:

Reviewer 1:

Comments and Suggestions for Authors

Q:The paper proposes Fusion High Resolution Network in diagnosing chest X ray images, which performs well compared to existing CNN methods. In general, the manuscript gives a good description of the proposed methodology and results and conclusions are supported by the empirical data analyzed in the study. However, I believe the manuscript needs to be revised. The quality of the collected data is very good but I think the analysis presented in the manuscript is somehow limited and needs to be extended. The introduction part should be improved so that the relation to "electronic" is highlighted.

Ans: Thanks for the reviewer's suggests. We have checked the full manuscript repeatedly and made improvement in the main body of the paper, especially in Result and Discussion section. For the results in Table 1, we detail the contents and significance of Table 1, highlight the average AUC and the increase rate of Mass. From Table 1, the horizontal and vertical comparison shows that our method achieves superior classification performance. Correspondingly, the theoretical content and experimental analysis will be extended. In Introduction part, we cite more references and elaborate our work’s relation to the "electronic".

Each comment has been addressed carefully in our revised manuscript and the modifications are highlighted in red.

Reviewer 2 Report

The authors developed a three-branch convolution neural network for diagnosing CXR images in their proposal. The proposed application is very interesting and important, because it may help the health personnel when making diagnose and reports about medical examinations like lung problems.

It seems that the FHRNe achieved the good AUC scores for the all the thorax diseases, unfortunately only 14 images, which represents a low number for testing an image classification algorithm. The authors must raise the number of images tested, when using their algorithm.

It seems that the Discussion section is very short (the worst part of the proposal), only 119 words. The authors should base their discussion on available published works made by other researchers. The discussion was made without presenting any confrontation with other works done by other  researchers in the field.

Finally, it is advised that the authors need to make a slight review/edit of the proposal for a more flow text and English grammar, because some parts of the text are not so easy to read.

Author Response

Journal: Electronics (ISSN 2079-9292)

Manuscript ID: electronics-678921

Type: Article   Number of Pages: 9

Title: Fusion High-Resolution Network for Diagnosing Chest X-ray Images

Dear Editor,

Thank you very much for your letter and for the comments by the reviewers. These comments are very valuable and helpful for our paper.

We appreciate the careful, constructive, and generally favorable reviews given to our paper by the reviewers.

We believe we have adequately addressed all the excellent advices and questions raised by reviewers. Furthermore, we checked the manuscript and made sure the submitted manuscript is correct.

Please contact us if any further questions remain.

Sincerely yours,

Prof.  Yu Pang 

Response to the comments of reviewers:

Reviewer 2:

Comments and Suggestions for Authors

Q:The authors developed a three-branch convolution neural network for diagnosing CXR images in their proposal. The proposed application is very interesting and important, because it may help the health personnel when making diagnose and reports about medical examinations like lung problems.

It seems that the FHRNe achieved the good AUC scores for the all the thorax diseases, unfortunately only 14 images, which represents a low number for testing an image classification algorithm. The authors must raise the number of images tested, when using their algorithm.

It seems that the Discussion section is very short (the worst part of the proposal), only 119 words. The authors should base their discussion on available published works made by other researchers. The discussion was made without presenting any confrontation with other works done by other researchers in the field.

Finally, it is advised that the authors need to make a slight review/edit of the proposal for a more flow text and English grammar, because some parts of the text are not so easy to read.

Ans: Thanks for the reviewer's suggests. According to your suggestion, we have read the article carefully and made some modifications accordingly.

Our modifications are described below:

In our experiment, we divided the ChestX-ray14 dataset into train set (total 75,708 images), validation set (total 10,816 images) and test set (total 25,596 images). The whole ChestX-ray14 dataset includes 14 different types of thorax disease images and 1 type of normal image. So, in Figure 3 the total number of image examples is 15. We use all 25,596 images to test our model, and prove the effectiveness of our algorithm. We have made improvement in Result and Discussion section. For the results in Table 1, we detail the contents and significance of Table 1, highlight the average AUC and the increase rate of Mass. Based on available published works made by other researchers we compare their results with ours in terms of AUC score and ROC curve. In Discussion part, we analyze our method can obtain satisfactory result on the basis of the fact that two significant structures are introduced, illustrate the effectiveness of the two structures through ablation experiments, and add Table 2 as the result of ablation experiments. It is precisely because our method introduces two uniquely innovative and important structures, which made our work without presenting any confrontation with other works done by other researchers in the field. Before being submitted, the manuscript has been edited for proper English language, grammar, punctuation, spelling, and overall style by one or more of the highly qualified native English-speaking editors at AJE. This document certify is shown below.

Maybe there is still little grammatical mistakes or partially not be flow. We will correct it as soon as we find it.

Round 2

Reviewer 1 Report

The manuscript focused on the fusion high-resolution network for diagnosing chest X-ray images. In general, the manuscript gives a good description of the proposed methodology in this study. Advancing the knowledge on these topics is relevant for the corresponding research community. Please find my comments below:

(1) The literature review is limited, because many approaches regarding the applications are not mentioned in the manuscript. Authors should consider citing "An automatic and intelligent optimal surface modeling method for composite tunnel structures" and "Multi-sensor technology for B-spline modelling and deformation analysis of composite structures" and so on.

(2) The manuscript is generally well readable but the references are inadequate which will affect the research significance of this manuscript.

Author Response

Journal: Electronics (ISSN 2079-9292)

Manuscript ID: electronics-678921

Type: Article

Number of Pages: 11

Title: Fusion High-Resolution Network for Diagnosing Chest X-ray Images

Dear Editor,

On behalf of all the authors of this paper, I would like to extend our sincere thanks to the editor and the anonymous reviewers for their great support and constructive comments, which provide us with momentum and guidance to make deeper research into our subject matter and further improve our paper. All the comments have been taken seriously and corresponding responses could be found in the revised edition. A detailed summary of changes is presented below.

Please contact us if any further questions remain.

Sincerely yours,

Prof.  Yu Pang

To Reviewer #1

Comments and Suggestions for Authors

The manuscript focused on the fusion high-resolution network for diagnosing chest X-ray images. In general, the manuscript gives a good description of the proposed methodology in this study. Advancing the knowledge on these topics is relevant for the corresponding research community. Please find my comments below:

(1) The literature review is limited, because many approaches regarding the applications are not mentioned in the manuscript. Authors should consider citing "An automatic and intelligent optimal surface modeling method for composite tunnel structures" and "Multi-sensor technology for B-spline modelling and deformation analysis of composite structures" and so on.

(2) The manuscript is generally well readable but the references are inadequate which will affect the research significance of this manuscript.

Ans: We are grateful for this comment. According to this comment, we have cited all the references suggested, with some corresponding changes in the text. Due to the similarities in research methods and their applications, we have added more related references on our method and other comparisons to make this manuscript more readable and significant.

Each comment has been addressed carefully in our revised manuscript, and the revised part is highlighted in red (in page 5 and 6) in re-submitted version.

Round 3

Reviewer 1 Report

The paper has been revised according to the comments and can be accepted in present form.

Author Response

Journal: Electronics (ISSN 2079-9292)
Manuscript ID: electronics-678921
Type: Article
Number of Pages: 12
Title: Fusion High-Resolution Network for Diagnosing Chest X-ray Images

Dear Editor,
Thank you very much for your comments. These comments are very valuable and helpful for our paper. All the comments have been taken seriously and corresponding responses could be found in the re-submitted edition. A detailed summary of changes is presented in our paper.(file name: electronics-678921-final)
The manuscript has been edited for proper English language and style by one or more of the highly qualified native English-speaking editors at AJE. (file name: electronics-678921-English.doc)
Please contact us if any further questions remain.
Sincerely yours,
Prof.  Yu Pang

To Reviewer #1
Comments and Suggestions for Authors
The paper has been revised according to the comments and can be accepted in present form.
Further Comments are provided below: 
"Very frequent interactions between the authors and a native speaker are necessary, as some descriptions are the one that only the authors know, and the native speaker needs to understand the contents of the paper. They are not just grammatical errors. Please instruct the proof reader and the authors about that, and force them to interact very closely."
Ans: According to the comments, the manuscript has been edited for proper English language, grammar, punctuation, spelling, and overall style by one or more of the highly qualified native English-speaking editors at AJE. This document certify is shown below.
